# The Role and Regulatory Mechanism of Brown Adipose Tissue Activation in Diet-Induced Thermogenesis in Health and Diseases

**DOI:** 10.3390/ijms23169448

**Published:** 2022-08-21

**Authors:** Pei-Chi Chan, Po-Shiuan Hsieh

**Affiliations:** 1Department of Physiology & Biophysics, National Defense Medical Center (NDMC), 161, Section 6 Min-Chuan East Road, Taipei 114, Taiwan; 2Graduate Institute of Medical Science, National Defense Medical Center (NDMC), 161, Section 6 Min-Chuan East Road, Taipei 114, Taiwan; 3Department of Medical Research, Tri-Service General Hospital, Taipei 114, Taiwan

**Keywords:** brown adipose tissue, diet-induced thermogenesis

## Abstract

Brown adipose tissue (BAT) has been considered a vital organ in response to non-shivering adaptive thermogenesis, which could be activated during cold exposure through the sympathetic nervous system (SNS) or under postprandial conditions contributing to diet-induced thermogenesis (DIT). Humans prefer to live within their thermal comfort or neutral zone with minimal energy expenditure created by wearing clothing, making shelters, or using an air conditioner to regulate their ambient temperature; thereby, DIT would become an important mechanism to counter-regulate energy intake and lipid accumulation. In addition, there has been a long interest in the intriguing possibility that a defect in DIT predisposes one to obesity and other metabolic diseases. Due to the recent advances in methodology to evaluate the functional activity of BAT and DIT, this updated review will focus on the role and regulatory mechanism of BAT biology in DIT in health and diseases and whether these mechanisms are applicable to humans.

## 1. Introduction

The thermogenic activity of BAT is stimulated by cold and also by a meal that induced a parallel increase in heat production, which has been well-documented to resist obesity through facilitating adaptive thermogenesis and energy expenditure (EE) both in rodents and humans [1]. BAT possesses a higher capacity to metabolize glucose and fatty acids, promotes heat production and EE, and is characterized by multilocular small lipid droplets, high mitochondrial density, and the expression of key thermogenic protein-uncoupling protein 1 (UCP1). There is evidence supporting that BAT could significantly affect whole-body energy metabolism in rodents and humans [2,3]. Therefore, BAT activation has been considered a potential therapeutic target in the treatment of obesity and related diseases [4].

In humans and rodents, the resting metabolic rate (RMR) accounts for 60–70% of whole-body EE, which is required for the performance of cellular and organ functions [5,6]. RMR is majorly determined by fat-free mass in the body. In general, gender difference and aging are the two main physiological factors to affect RMR [7,8,9], which may be due to differences in fat-free mass, especially skeletal muscle [10]. DIT is the increase in EE associated with food intake [11] and accounts for 5–15% of total EE [12]. The magnitude of the thermic effect of food depends on diet composition and caloric intake. DIT represents about 10% of daily total EE in healthy subjects [11]. DIT also could be affected by oral stimuli (i.e., the duration of tasting food and chewing in the mouth) [13], and environmental factors [14].

## 2. Translational Aspects of BAT-Mediated Adaptive Thermogenesis on the Maintenance of Whole-Body EE

Obesity caused by excess fat storage is a consequence of a long-term imbalance of energy intake and EE, which is detrimental to the health of people. Understanding the counter-regulatory mechanisms of energy metabolism is critical to the development of effective therapies to treat the epidemic of obesity. However, so far, therapeutic approaches aiming at increasing EE remain inconclusive. The currently developed drug targets have either low efficacy or severe cardiometabolic effects [15], which remain crucial for future drug development [4,16].

BAT has been speculated as a potential anti-obesity target due to its capacity for adaptive thermogenic activity and contribution to total daily EE. So far, the studies conducted with rodent models have provided a major understanding of the BAT contribution to whole-body energy metabolism [17]. Notwithstanding, the majority of fat depots are similar in rodents and humans. Mouse BAT is mainly distributed in the interscapular BAT and some small depots are found in the cervical, axillary, perivascular, and perirenal regions [18]. Metabolically active BAT is found in adult humans in the neck and supraclavicular regions [18,19].

A recent human study has demonstrated that subjects with BAT activity were independently correlated with a lower prevalence of type 2 diabetes, dyslipidemia, and cardiovascular diseases. In particular, the beneficial effects were more pronounced in overweight or obese individuals with BAT, indicating that BAT might play a role in mitigating obesity and its comorbidities [20].

## 3. The Dual Role of BAT Activation under the States of Energy Needed and Energy Excess Conditions

The balance of energy intake and EE is essential for the maintenance of energy homeostasis and body weight. Total EE is made up of the sum of energy (calories) used to maintain the cells and organ function on a daily basis, and is composed of five main components: (1) RMR, (2) activity-related EE (AEE), (3) thermic effect of food (TEF), (4) adaptive thermogenesis, and (5) growth [5,21]. BAT activation is involved in the mechanism underlying at least three major components of total EE (Item 2, 3, 4), indicating its importance in the regulation of energy homeostasis. In addition, adaptive thermogenesis including shivering and non-shivering thermogenesis is crucial to maintaining mammal body temperature [22,23]. In non-shivering thermogenesis, brown and brite/beige fat dissipates chemical energy as heat by UCP1-dependent thermogenesis [24], as well as by other “futile” enzymatic cycles UCP1 independent thermogenesis [24,25].

Cold exposure is the most effective physiological regimen to activate BAT. However, it would be hard to increase human exposure to cold temperatures under well-controlled ambient temperature with the presence of clothing and an air conditioner. On the other hand, it remains uncertain whether the regulatory signal pathways of cold-induced thermogenesis (CIT) could be the potential drug targets for anti-obesity treatment due to adverse effects of chronic cold exposure such as elevated blood pressure [15] and deterioration of the development of atherosclerosis [26].

On the other hand, the study of Rothwell and Stock [27] in 1979 conducted with male Sprague-Dawley rats demonstrated that rats with high-fat and cafeteria diet feeding exhibited less weight gain than expected based on energy intake with a simultaneous increase in BAT activity and EE. In addition, a previous study has shown that BAT’s thermogenic response to diet feeding is not observed in animals without UCP1 [28]. Thus, the role of UCP1-dependent thermogenesis and EE in BAT in the maintenance of energy balance after a meal seems to have been noted [29]. Moreover, DIT is the thermogenesis generated after meals and is another component of non-shivering thermogenesis. The involvement of BAT in DIT has been demonstrated in small rodents recently [30,31]. There was also some updated information available on humans [3,32].

## 4. Diet-Induced Activation of BAT

### 4.1. Definition of DIT

DIT is defined as an increase in EE after a meal above that of the fasting state and is related to digestion, intestinal absorption, and storage of these nutrients, contributing 5–15% to total daily EE. DIT is generally composed of two components: an obligatory part and a facultative part. Obligatory DIT is also called “the thermic effect of food”, which refers to the heat necessarily released during digestion and processing of the food. The values for the obligatory DIT are for carbohydrates 5–10%, lipids 0–3%, and protein 20–30% [33]. The current methods to obtain the above values cannot be excluded since the thermogenesis is parallelly generated from facultative DIT. For instance, DIT in humans is generally measured using ventilated hood systems [34,35,36] or respiratory chambers [37,38,39,40]. In mice, DIT has been estimated by measuring the postprandial increase in body temperature and the increase in oxygen consumption (VO_2_) [31,41,42,43]. Although the existence of facultative DIT and the role of BAT for DIT remains debated [29,32,44], previous studies showed that facultative DIT was mediated, at least partly, through BAT-mediated thermogenesis [27,45]. Several reports have also shown that DIT can increase EE in response to overfeeding and eventually, retard excessive energy intake and weight gain in rodents [27,46,47,48]. In adult humans, several studies have indicated that the activation of BAT is involved in the generation of DIT [49,50,51]. Thus, the involvement of BAT activation in DIT has been considered the potential target to retard the development of overweight and obesity [27,52].

### 4.2. The Role of BAT Activation in Facultative Adaptive DIT and Whole-Body EE

BAT is one of the main sites for the generation of DIT. Recent studies have reported that the activation state of BAT is increased in animals fed energy-rich diets [27,28,30,52,53,54,55,56] and significantly contributes to the control of energy metabolism. On the other hand, DIT is higher in BAT-positive male subjects measured by ^18^F-fluorodeoxyglucose (^18^F-FDG)-PET/CT which could explain the lower incidence of obesity, insulin resistance, and diabetes mellitus type 2 than BAT-negative subjects [51]. Moreover, it has been reported that activation of human BAT during DIT is correlated with body mass index (BMI), sex, and age [57,58]. In addition, the augmentation of oxygen consumption, respiratory enzyme activity, and UCP1 protein, in parallel with an increased BAT activity and EE were noted in rats fed high fat and cafeteria diets [27,42]. On the other hand, whole-body oxygen consumption after a meal in UCP1-deficient mice was lower than that in wild-type control. Accordingly, UCP1-deficient mice are prone to develop obesity and insulin resistance when mice are fed a high-fat diet (HFD) and maintained at a thermoneutral temperature (30 °C). It was suggested that DIT could be induced by adrenergic stimulation of BAT activity through activation of the BAT-specific UCP1 protein [28,52]. Accordingly, UCP1-mediated thermogenesis is important for DIT in humans, and the possible contribution of BAT thermogenesis to DIT and regulation of energy balance have been suggested by the study focusing on single nucleotide polymorphism in the UCP1 gene. To examine the effects of A-3826G mutation in the UCP1 gene on DIT, Nagai et al. demonstrated that this UCP1 gene mutation could reduce postprandial thermogenesis in response to a high-fat meal shown in healthy boys [59].

On the other hand, it has been reported that the inactivation of UCP1 did not potentiate diet-induced obesity in mice. Recent studies have demonstrated that multiple UCP1 independent thermogenic mechanisms such as Ca^2+^ futile cycling through the SERCA2b-RyR2 pathway in the endoplasmic reticulum, creatine cycling in the mitochondria, and lipolysis/re-esterification (to provide fuel for futile cycling based on ATP sinks centered on fatty acid-mediated leak pathways driven by the ADP/ATP carrier) [24,25]. UCP1-independent EE has begun to be therapeutic potential that lies in refining our understanding of the regulation of adaptive thermogenesis. Accordingly, Anunciado-Koza and colleagues have demonstrated that the increase in oxygen consumption in UCP1^+/+^ and UCP1^−/−^ mice were comparable when the diet is switched from chow to high fat or sucrose [29], indicating that DIT might be generated through both UCP1 dependent and independent pathways. Taken together, the current observation implicates that the thermogenic activation of BAT is involved in the development of DIT both in human and animal studies.

## 5. Mechanisms of DIT

BAT thermogenic activation significantly contributes to the DIT in postprandial conditions [32]. To date, there are many potential factors and mechanisms have been reported to participate in DIT. As shown in Figure 1, we summarized the updated reports about the regulatory mechanisms involved in diet-induced BAT activation (DIT) and the role of BAT activation in DIT in health and diseases. However, the precise nature of their roles and regulatory mechanisms of BAT activation in DIT and whole-body EE remains largely unexplained.

### 5.1. Sympathetic Nervous System (SNS)–BAT Axis

CIT could be generated by activating SNS and inducing BAT thermogenesis through the β3-adrenergic receptor (ADRB3) in mice and β2-adrenergic receptor in humans [60]. Based on the principal role of the SNS-βAR axis for CIT, it is conceivable that this axis might also be a key mechanism in DIT. For instance, SNS activity in BAT calculated from tissue norepinephrine turnover rate is increased in mice with long-term cafeteria and high-calorie diet feeding [61,62]. In addition, studies [63] also showed that surgical denervation of BAT could significantly attenuate metabolic activation of BAT after intake of a liquid meal in rats. Accordingly, denervation of BAT also alleviates the increase in mitochondrial GDP binding, total UCP1 protein content, and mitochondrial content in rats on an energy-enriched diet [64]. These results support the content that BAT activation in DIT is, at least in part, mediated through SNS activation. Moreover, previous reports have further demonstrated that food palatability and oropharyngeal taste sensation are also substantially involved in diet-induced sympathetic activation and BAT thermogenesis [65,66,67].

### 5.2. Gut-Secreted Peptides and Hormones and Bile Acids

Since the release of gastrointestinal peptides is one of the acute physiological responses to eating, some of these peptides have been reported to act as mediators to regulate DIT. For instance, gut hormones could activate BAT through their effects on the efferent SNS tone, such as cholecystokinin (CCK) [68,69] and glucagon-like peptide (GLP-1) [70]. In addition, the study of Li et al. [71] revealed a novel endocrine gut–BAT–brain axis triggered by secretin that initiates the canonical secretin receptor (SCTR)–cAMP–PKA–lipolysis–UCP1 pathway in brown adipocytes from the intestine during a meal. Their observation supported that a postprandial increase in circulating secretin activates BAT thermogenesis by binding to SCTR in brown adipocytes. On the other hand, bile acids have been documented to activate BAT directly through Takeda G-protein receptor 5 (TGR5) in brown adipocytes [72]. Watanabe et al., [73] further demonstrated that bile acids activate TGR5 in mouse brown adipocytes and then, facilitate thermogenic activity through type 2 deiodinase (D2) activation. Accordingly, direct stimulatory effects of bile acids (chenodeoxycholic acid) on BAT activity have been evidenced in humans using brown adipocytes in vitro and using FDG-PET/CT scan in vivo [73,74]. TGR5 has also been demonstrated to participate in browning of white adipose tissue under cold exposure and prolonged HFD feeding [75]. Thereby, targeting TGR5–BAT axis with bile acids or drugs could be a promising target for combating obesity and related metabolic disorders in humans.

### 5.3. Insulin

It is well recognized that the role of insulin as a key hormone in the storage of ingested nutrients, and also possesses the capacity to modulate energy balance after a meal. A recent study has indicated that insulin is a crucial protein involved in the mitochondrial bioenergetic and thermogenic capacity of brown adipocytes [76]. In fact, BAT appears to be differently activated by insulin and cold. When activated by cold, it dissipates energy in a perfusion-dependent manner. Nevertheless, in response to insulin, BAT glucose uptake could have a 5-fold increase independently of its perfusion [77,78]. In addition, it is suggested that insulin may play an important role in the dietary induction of facilitative adaptive thermogenesis. For instance, the study conducted with ten healthy lean volunteers and a euglycemic clamp method in conjunction with respiratory exchange measurements, has shown the progressive increase in RMR along with the increase in the glucose infusion rate without changes in insulin and norepinephrine concentrations. It is implicated that insulin-stimulated glucose uptake is crucial for the thermogenic response of insulin [79]. On the other hand, Lee et al. have demonstrated that the recruitment of human BAT is accompanied by augmented DIT and postprandial insulin sensitivity [80]. Thus, it appears that the interactions between insulin and thermogenesis seen in rats could also exist in humans after meals.

Insulin resistance is a well-characterized consequence of obesity. Cafeteria diet feeding was conducted with normal and diabetic animals to assess changes in insulin-mediated glucose metabolism and to investigate the effects of insulin resistance on the development of DIT [81]. The attenuated DIT shown in cafeteria-fed diabetic rats was restored by an intensive program of insulin administration [81]. In addition, insulin resistance could result in a defect in insulin-mediated thermogenic response which contributes to the pathogenesis of obesity in humans [80]. The study of Aherne and Hull [82] conducted with human autopsy demonstrated that for obese infants of diabetic mothers exhibited impaired BAT function, further suggesting that maternal insulin deficiency may affect the thermogenic function of BAT in the offspring. Nevertheless, a recent report from Loeliger, R.C. has shown that DIT was not associated with BAT activity. DIT after an oral glucose load was not associated with stimulated ^18^F-FDG uptake into BAT, suggesting that DIT is independent of BAT activity caused by facilitating glucose uptake in humans [32]. The underlying mechanism of insulin on DIT is needed to be further clarified.

### 5.4. Liver-Derived Factors

Hepatic tissue has been documented to be involved to a significant extent in the thermogenesis process. The study was conducted with rats fed chow or a cafeteria diet of highly palatable human foods and measured the RMRs (VO_2_) before and shortly after two-thirds hepatectomy or sham operation at thermoneutrality (28 °C), and again after administration of propranolol (5 mg/kg). The difference in RMR between the resting period and propranolol treated period was used to represent the DIT. Their result suggests that liver is the major (70–100%) effector of the DIT [83,84]. In addition, Wallace et al. have demonstrated for the first time that the enhancement of specific calorigenic metabolic processes such as mitochondrial glycerophosphate oxidase within the liver contributes significantly to the heat production of DIT in cafeteria-fed rats [85]. These observations implicate that hepatic tissue might substantially take part in DIT under postprandial conditions.

### 5.5. Leptin

The adipocyte-derived leptin is a key factor in the regulation of energy balance [86], which is significantly increased in the obese state [87]. In fact, leptin has been reported to activate BAT sympathetic activity [88] and heat production [89,90,91]. In addition, a recent report has shown that refeeding increased plasma leptin concentrations in 48-h-fasted lean rats and leptin could mediate the increases in body temperature through hypothalamus-adrenal medulla-adipose tissue crosstalk after a meal [92]. Accordingly, chronic leptin treatment could stimulate BAT-mediated thermogenesis in leptin-deficient *ob/ob* mice by enhancing sympathetic innervation and activation [93]. Furthermore, BAT-ectomized rats reduced the thermogenic effect of food by 60% [92], suggesting that BAT appears to be a crucial mediator of leptin-induced thermogenesis. However, some earlier studies have found that female leptin-deficient *ob/ob* mice fed a palatable cafeteria diet still result in increased leptin-independent SNS activity and thermogenesis in BAT [94]. Some other reports also pointed out that leptin do not possess thermogenic function [95,96] and act as the mediator of DIT. These controversial observations might attribute to the differences in gender, methods, and experimental condition, which are needed to be further elucidated.

In contrast to leptin, adiponectin exhibits several characteristics of inhibiting energy expenditure. For instance, adiponectin inhibits UCP1 expression by reducing ADRB3 expression in brown adipocytes, along with reducing BAT thermogenesis in mice [97]. Previous report showed that neither fasting nor refeeding changed adiponectin serum levels in both young and old male rats [98]. The involvement of adiponectin in DIT process remains ambiguous.

### 5.6. Muscle

Sarcolipin (SLN) is a regulator of the Sarco/endoplasmic reticulum Ca^2+^-ATPase (SERCA) pump in muscle has been proposed to play an important role in DIT [55,99,100,101] and EE [102]. For instance, SLN regulates muscle-based non-shivering thermogenesis and is up-regulated with HFD. In addition, SLN KO mice are prone to develop diet-induced obesity and glucose intolerance. On the other hand, mice with SLN gene deletion gained comparable weight as UCP1-deficient mice on HFD, implicating that loss of muscle-based thermogenesis has similar consequences on weight gain as loss of UCP1-mediated thermogenesis [100]. Moreover, the body weight of mice with SLN overexpression is significantly lower than that of littermate control and they were resistant to HFD-induced obesity [101]. These mechanistic experiments indicated that SLN as an uncoupler of SERCA pump might create both energy demand and increase energy expenditure and are essential for counter-regulation of diet-induced energy intake.

### 5.7. Thyroid Hormones

Thyroid hormones, thyroxine (T4) and triiodothyronine (T3), have been well-documented to be a crucial determinant of energy expenditure and basal metabolic rate and also involved in the regulation of thermogenic responses to cold and diet [103]. Both cold-challenged and cafeteria-fed rats exhibited increases in circulating thyroid hormone levels [104]. In rats, overnutrition induced by a cafeteria diet is accompanied by a large increase in EE along with plasma T3 concentration [105]. T3 also acts as the main regulator of producing DIT in birds [106]. Accordingly, long-term over-feeding in men also exhibited accelerated metabolic rate and an increase in circulating T3 level [107]. Taken together, these observations implicate that thyroid hormone, especially T3, might substantially contribute to the generation of DIT under postprandial condition.

## 6. Food-Derived Stimulants and Diet-Induced BAT Thermogenesis

Dietary molecules are important factors in potentiating BAT thermogenic functions and affecting energy metabolism. Nutritional modulation of adaptive thermogenesis is a newly developing research field and of clinical importance to evaluate its potential as a therapeutic target to prevent the development of obesity and promote human health. There are several signaling pathways that have been reported to be involved in stimulating BAT activation by dietary molecules.

### 6.1. Caloric Intake and Circadian Rhythmicity

An increase in calorie intake or meal size has been demonstrated to facilitate DIT [6,108,109]. In contrast, caloric restriction leads to reduce hormonal responses for insulin and incretin (GIP and GLP-1) and lower DIT in healthy adults [109]. On the other hand, the extremely hypocaloric diet (200 kcal) could induce similar DIT to the moderately hypocaloric diet (400 kcal), which led to a negative energy balance and greater thermogenic efficiency in healthy male adults [109]. In addition, a study regarding circadian rhythmicity in the human DIT showed that after a hypocaloric diet (1046 kcal/day) subjects displayed more efficient energetic value in the morning than in the evening, implicating that the timing of food intake is also important for successful weight loss under conditions of caloric restriction [110]. Thus, these studies suggest that caloric intake and circadian rhythmicity are substantially involved in the regulation of postprandial energy metabolism.

### 6.2. Phytochemicals (e.g., Capsaicin, Resveratrol, Curcumin, and Green Tea)

The effects of capsaicin, the major pungent component of chili peppers, and its analogs capsinoids on BAT-mediated thermogenesis and EE have been intensively studied. On the other hand, previous studies have demonstrated that chemical activation of TRP channels by food ingredients can recruit and activate BAT in humans and animal models [111,112,113]. The transient receptor potential vanilloid (TRPV)1 located on primary afferent neurons has been speculated to be a primary target for capsinoids [114]. Both capsaicin and capsinoids are potent in activating TRPV1 expression in sensory nerves within the gastrointestinal tract and increasing SNS activity in BAT. Thereafter, they would induce a rapid increase in BAT temperature and subsequently facilitate whole-body EE and fat oxidation in small rodents [115] and humans [116,117,118].

Resveratrol (3,5,4-trihydroxystilbene), a natural polyphenolic compound found in the skin of grapes and other plants, has been demonstrated to prevent obesity and related complications [119,120]. Resveratrol has also been suggested to facilitate thermogenesis in BAT [121]. Resveratrol could induce UCP1 gene expression and oxygen consumption in BAT of mice fed either normal chow or HFD [122]. Accordingly, in animal models of obesity, the addition of resveratrol could reverse HFD-induced suppression of UCP1 expression and oxygen consumption in BAT, resulting in an increased basic metabolic rate [123,124]. Furthermore, there is compelling evidence that resveratrol promotes BAT activation through distinct molecular pathways via synergistic mechanisms of mitochondrial biogenesis, AMP-kinase signaling, SIRT1 activity, and cAMP signaling [121,123,125,126].

Curcumin is the principal curcuminoid of Indian spice turmeric, which is a member of the ginger family. Curcumin has been shown to have a protective effect on obesity and metabolic diseases [127,128,129]. The anti-obesity effects of curcumin were not exhibited in UCP1-deficient mice [130], implicating the involvement of UCP1-dependent thermogenesis. A recent report further suggests that gut microbiota mediates the effects of curcumin on enhancing UCP1-dependent thermogenesis and improving high-fat diet-induced obesity in mice [130]. Curcumin has been reported to enhance EE through stimulating thermogenesis by enhancing the expression of AMPK, SIRT1, proliferator-activated-receptor-gamma-coactivator1-α (PGC-1α), catechol-O-methyltransferase (COMT) and SNS activity in BAT [131,132,133]. In addition, curcumin also increased the expression of critical mitochondrial proteins such as carnitine palmitoyltransferase I (CPT1, responsible for the oxidation of fatty acids) and cytochrome C (responsible for mitochondrial oxidative phosphorylation) as well as the number of mitochondria in brown adipocytes [134].

The consumption of green tea or its components, green tea catechins, have been reported to be able to increase fat oxidation and energy expenditure along with upregulated UCP1 expression in rat BAT, suggesting that suppressive effect of catechins on body fat accumulation is associated with increased BAT activity [135]. Catechins in green tea could facilitate noradrenaline and ADRB3 signaling activation and subsequently stimulate thermogenesis and fat oxidation in mice [136,137]. However, thermogenic activation by green tea catechins remains controversial in humans. In obese men, consuming tea catechins, such as EGCG increased postprandial fat oxidation, but failed to change energy expenditure [138]. In addition, green tea supplementation has been reported to upregulate uncoupling protein 3 expression in adipose tissue of severely obese women without promoting weight loss [139]. On the other hand, a meta-analysis of placebo-controlled studies showed that intake of catechin-caffeine mixtures facilitated energy expenditure and fat oxidation in humans [140]. Notedly, catechin treatment might cause adverse side effects, especially in cardiovascular systems through stimulation of the SNS activity in long-term usage.

### 6.3. Dietary Fatty Acids

Fish oil enriched with n-3 polyunsaturated fatty acids, eicosapentaenoic acid (EPA), and docosahexaenoic acid (DHA), has an anti-obesity effect in humans and rodents [141,142,143,144]. Fish oil could also promote thermogenesis and energy expenditure accompanied by augmented UCP1 expression in brown/beige adipocytes [145]. The effect of fish oil was likely mediated by the SNS and TRPV1, leading to ADRB3 stimulation and BAT thermogenesis in mice. EPA-mediated browning effects were abolished in Trpv1 knockout mice [146]. Moreover, dietary fish oil has been reported to increase DIT in mice via the increased UCP1 activation in BAT and induced beige biogenesis in subcutaneous WAT. In addition, these fish oil-induced mechanisms in BAT contribute to activate PPARα via the binding to peroxisome proliferator-activated receptor (PPAR) α [147]. Peroxisome proliferator-activated receptors (PPARs) are members of a nuclear receptor superfamily involved in the regulation of nutrient-dependent transcription. Peroxisome-derived lipids has been reported to regulate adipose thermogenesis by mediating cold-induced mitochondrial fission [148]. On the other hand, pemafibrate, a selective PPARα agonist, activates thermogenesis in subcutaneous (inguinal) white and brown adipose tissues by increasing plasma levels of fibroblast growth factor 21 and protecting again diet-induced obesity in mice [149]. These observations suggest that fish oil might be a promising thermogenic dietary fat to against obesity [150].

### 6.4. All-Trans Retinoic Acid and a Vitamin A Metabolite

Retinoic acid is an active metabolite of vitamin A and has been reported to substantially contribute to BAT thermogenesis via nuclear retinoic acid receptors (RARs). In vitro experiment, retinoic acid has been reported to induce expression of UCP1 in mouse beige and brown adipocytes [151] and facilitate BAT activation via unique transcriptional regulation, instead of ADRB3-mediated signaling. Retinoic acid-treated animals have shown increases in UCP1 expression and improved RAR and PPARδ activation, leading to augmented BAT activation and fatty acid oxidation [152]. Conversely, a vitamin A–deficient diet feeding increased body weight, whole body fat mass, and reduced BAT thermogenic potential in mice [153]. Accordingly, a study in healthy adults showed the inverse relationship between vitamin A intake and adiposity [154]. Additionally, in morbidly obese subjects, it exhibited the inverse correlation between serum retinol and BMI [155]. It is implicated that dietary supplement of retinoic acid possesses anti-obesity potential via promoting DIT-associated thermogenesis and whole-body EE.

## 7. BAT Dysfunction in Metabolic Diseases

### 7.1. Obesity

Obesity is characterized by excessive fat storage due to long-term energy intake over EE [156]. DIT can be an important component in regulating whole-body EE and there has been speculated that a defect in DIT predisposes to obesity [14,157,158]. The thermogenic response to food has been proposed to act as a causative factor for obesity by de Jonge and Bray [159]. They reported in their review that 29 of 49 relevant studies found the thermic effects of food to be lower in obese than in lean subjects. Animal studies for DIT also provided evidence that obesity phenotypes in *ob/ob* mice were associated with defective BAT adaptive thermogenesis [160]. Thus, it is implicated that obesity might be causally linked to a defective BAT in an obese animal model. Nevertheless, it has been reported that energy expenditure positively correlates with the increase in body mass in humans. Accordingly, severe obesity is associated with an increase in EE [161]. These observations indicate that the regulatory mechanism of EE in obese mice and obese subjects might be different. Nevertheless, the association between lower DIT and obesity also remains controversial. Some reports have demonstrated that no difference exists in DIT between obese and nonobese individuals [108,162,163,164,165]. For instance, Tentolouris et al. [163] have shown that there are no differences in DIT or macronutrient oxidation between nonobese and obese subjects after a carbohydrate-rich meal and a fat-rich meal. D’Alessio et al. [108] tested the effect of four mixed diets containing different amounts of fat-free calories (kcal/kg) intake. Their result showed the increase in EE in the obese and nonobese groups were not different. However, the other report has shown a greater increase in EE in obese European men than in their nonobese counterparts after consuming an isocaloric test meal [164]. Since DIT is influenced by the amount of caloric intake and diet macronutrient composition [110], it would be crucial to be standardized for both the caloric and composition content of the meal, while comparing DIT between obese and nonobese individuals. Therefore, more studies are needed to elucidate the role and the mechanism underlying DIT in obese and nonobese subjects, which is of clinical importance for developing potential anti-obesity diet supplements.

### 7.2. Diabetes

The obesity-associated insulin resistance is a crucial risk factor in the development of type 2 diabetes. On the other hand, the development of type 2 diabetes exhibits not only the derangement in glucose disposal (e.g., insulin resistance) but also a defect in thermogenic function, which would result in an increase in weight gain [166]. Rothwell and Stock [64] have reported that streptozotocin-induced diabetes (defect of insulin secretion) abolishes the increase in metabolic rate associated with cafeteria feeding. Impaired DIT in cafeteria-fed diabetic rats could be recovered by insulin administration [81]. Furthermore, the study conducted with type 2 diabetic patients showed that there is a diminution in thermogenic response to intravenous insulin infusion along with diminished insulin-mediated glucose uptake [79]. Accordingly, glucose-induced postprandial thermogenesis is decreased in the presence of insulin resistance and/or reduced insulin response to the glucose load in nondiabetic and diabetic obese subjects [167].

### 7.3. Hyperlipidemia

Hyperlipidemic patients with obesity and type 2 diabetes characterized by high plasma TG and low plasma HDL-cholesterol concentrations have been reported to exhibit lower thermogenic adipocyte activity than that in healthy subjects [4,168]. In addition, it has been demonstrated that the expression of UCP1 in human epicardial fat is positively correlated with lower plasma TG and higher plasma HDL-cholesterol levels [169]. BAT activity is not only attenuated in hyperlipidemia with obesity but also inversely correlated with BMI and body fat mass. On the other hand, since thermogenesis of BAT consumes large amounts of fatty acids, BAT has been considered to play a key role in TG clearance [170]. Brown adipocytes take up fatty acids via CD36 and oxidize them to fuel thermogenesis, while the liver clears plasma LDL remnants [170]. Ablation of BAT could result in hyperlipidemia in mice [171], indicating that BAT plays a critical role in maintaining TG homeostasis in circulation. Moreover, BAT activation has been documented to lead to the decomposition of triglycerides in cells through increasing oxidative metabolism and UCP1-dependent thermogenesis [3]. Thereby, BAT activity might be causally involved in the pathogenesis of hyperlipidemia in human subjects.

Nevertheless, it has been discovered that decreased activity of BAT is associated with insulin resistance, which can alter systemic lipid metabolism, consequently leading to the development of hyperlipidemia [172]. Thereby, insulin resistance could also be an important confounding factor to affect DIT in hyperlipidemic subjects.

### 7.4. NAFLD

NAFLD is highly associated with obesity and related metabolic disorders. Several environmental factors are known to affect NAFLD including a Western diet, high fat and cholesterol-enriched diet, and a high fructose-enriched diet [173]. Insulin resistance is also a key factor in the pathogenesis of NAFLD. In a clinical study, individuals with NAFLD have lower BAT activity than counterpart control and BAT activity is inversely related to insulin resistance and hepatic fat accumulation [174]. On the other hand, BAT has been reported to possess the capability to protect against NAFLD by facilitating the utilization of FFAs to fuel its thermogenesis and activation of BAT and subsequently, decreasing both circulating FFA levels and uptake of FFAs into the liver [174]. In preclinical studies, enhancing the metabolic activity of BAT potently reduces liver lipid accumulation in rodents [175]. BAT activity in humans is also associated with reduced risk for NAFLD [176], suggesting that BAT activation might play a protective role in human NAFLD. Croci et al., conducted with 20 patients with NAFLD showed that the low EE in NAFLD subjects was mainly due to lower fat-oxidation in basal conditions than in healthy control [177]. These observations have demonstrated that the anti-liver steatosis function of BAT is likely to be conserved both in rodents and humans.

### 7.5. Cancer-Associated Cachexia (CAC)

BAT is not only involved in the regulation of adaptive thermogenesis and energy balance but also contributes to the physiologic perturbations associated with cachexia [178]. The activation of BAT and augmented UCP1 expression were noted in the development of cancer cachexia. Moreover, inflammatory signaling as an energetically wasteful and maladaptive response to anorexia during the development of cachexia has been considered to play an important role in the modification of BAT biology [178]. CAC could cause profound metabolic disturbance and is characterized by progressing muscle wasting and adipose loss and asthenia. CAC is not solely attributable to inadequate nutritional intake cause severely increases the morbidity and mortality risk in cancer patients [179]. Weight loss has been ascribed to a mechanism of altered DIT which leads to a reduction in the efficiency with which ingested substrate is stored [180]. However, it has been demonstrated that the reduction of DIT in cancer patients is another element of starvation adaptation contributing to their weight loss. It is implicated that altered thermogenesis does not contribute to the weight loss seen in cancer cachexia.

## 8. Conclusions and Future Perspectives

BAT has been speculated as a promising target for combating obesity and metabolic diseases. Apart from CIT, BAT has been suggested to play a facilitative role in the generation of DIT. Diet-induced BAT activation may contribute more significantly to the regulation of whole-body energy metabolism and weight control in our usual daily life. Benefit from recent advances in technologies to evaluate the diet-induced BAT activation, many exogenous and endogenous regulatory factors such as diet nutrient components, thermogenic food ingredients, various postprandially secreted humoral factors and SNS activity have been reported to modulate BAT thermogenesis and EE through multiple thermogenesis-related signal pathways as shown in Table 1. This review provides updated information regarding the role of BAT activation in DIT and the possible regulatory mechanisms. It would help to facilitate the translation development of potential dietary factors/food ingredients that can trigger diet-induced BAT activation and possess anti-obesity potential. On the other hand, it would help to understand the pathological link between BAT activation in DIT and the development of metabolic diseases.

## Author Contributions

P.-C.C. developed the first manuscript draft; P.-S.H. reviewed, revised, and approved the final version of the manuscript. All authors have read and agreed to the published version of the manuscript.

## Figures and Tables

**Figure 1 ijms-23-09448-f001:**
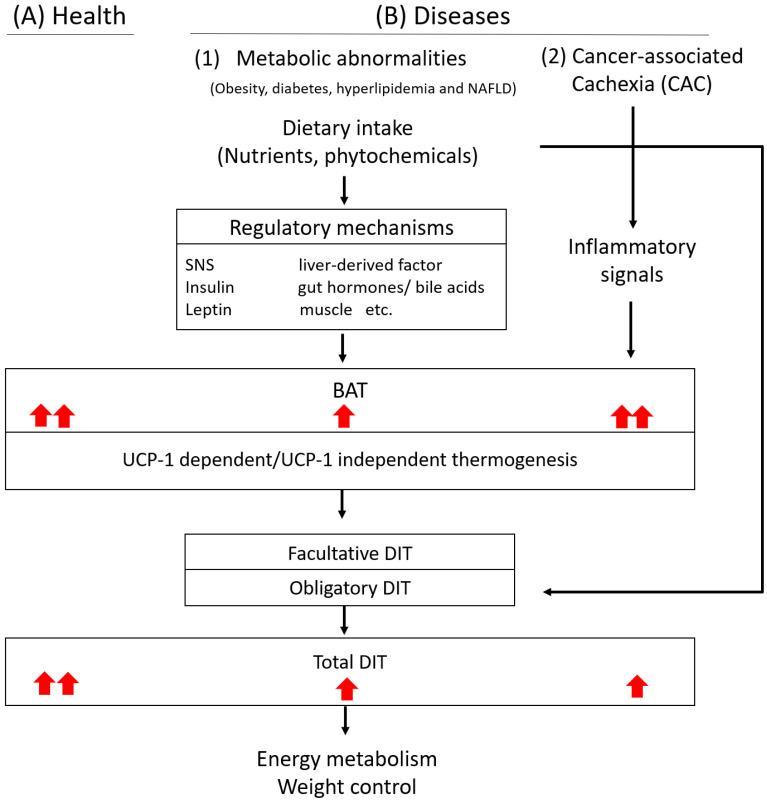
The role and regulatory mechanism of BAT activation in DIT in health and diseases. (**A**) Under healthy condition, dietary intake could promote BAT activation (facultative DIT) and total DIT to facilitate whole-body energy metabolism. (**B**) Under disease conditions: (1) Metabolic abnormalities such as obesity, diabetes, hyperlipidemia and NAFLD, dietary intake-induced BAT activation, and total DIT will be attenuated and might subsequently, affect whole-body energy balance and weight control. (2) Cancer-associated Cachexia (CAC) is characterized by systemic inflammation, which could enhance thermogenesis in BAT and increase systemic energy expenditure. However, low DIT were noted in the development of cancer cachexia.

**Table 1 ijms-23-09448-t001:** The summary of the animal and human studies about BAT activity and DIT in health and diseases.

**DIT in health**							
**Intervention**	**BAT**		**DIT**	**Total EE**	**Energy** **Balance**	**Experimental Model**	**Reference**
	**Mechanism of Action**	**Activity**					
Caloric intake							
low			↑	↑		humans	[108,109]
high			↑↑	↑↑		humans	[107,108]
Circadian rhythmicity							
day			↑↑	↑↑		humans	[109]
night			↑	↑		humans	[109]
Capsaicin	↑ ADRB3	↑↑	↑↑	↑↑	一	mice	[115]
	↑ TRPV					humans	[116,117,118]
Resveratrol	↑ AMPK	↑↑	↑↑	↑↑	一	mice	[122,124,125]
	↑ SIRT1					humans	[126]
Curcumin	↑ ADRB3	↑↑	↑↑	↑↑	一	rats	[129]
	↑ cAMP/PKA					mice	[130]
	↑ AMPK					cells	[131]
Green tea	↑ ADRB3	↑↑	↑↑		±	humans	[138,139,140]
Fish oil		↑↑	↑↑	↑↑	±	humans	[145]
	↑ TRPV				一	mice	[146]
all-trans retinoic acid	↑ PPARγ	↑↑	↑↑	↑↑		cells	[140]
						mice	[141,142]
**DIT in diseases**							
**Intervention**	**BAT**		**DIT**	**Total EE**	**Energy** **Balance**	**Experimental Model**	**Reference**
	**Mechanism of Action**	**Activity**					
Metabolic abnormalities							
Obesity			↑			humans	[159]
			↑			mice	[160]
T2DM			↑	↑		rats	[64,81]
		↑	↑			humans	[79,167]
Hyperlipidemia		↑		↑		humans	[170]
						mice	[171]
NAFLD		↑				mice	[174]
				↑		humans	[174,176,177]
Cancer-associated cachexia (CAC)		↑↑				mice	[178,179]
			↑	↑	一	humans	[180]

## Data Availability

All the data are presented in the manuscript.

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
