# Peer review of "The Role and Regulatory Mechanism of Brown Adipose Tissue Activation in Diet-Induced Thermogenesis in Health and Diseases"

_ijms, 2022, doi:10.3390/ijms23169448_

Round 1

Reviewer 1 Report

Authors proposed in this paper to review the regulatory mechanisms underlying diet-induced thermogenesis (DIT) in brown adipose tissue (BAT). This interesting review discussed the implication of BAT in the body energy expenditure and  the role of DIT in the activation of BAT thermogenesis. Different aspects of particular tissues, as liver and muscle, and their interactions with key hormones regulating the whole metabolism were well exposed. The role of several food micronutrients in DIT was highlighted.

However, several point should be addressed:

-       The role of peroxisome proliferation and peroxisomal fatty acid beta-oxidation are well known to play an important role in thermogenesis. Knowing that peroxisomal fatty acid oxidation produce heat and not ATP and this pathway is dependent on nutritional conditions, authors could also discuss this point.

-       Several abbreviations are not explained: UCP1, F-FDG-PET/CT..etc

-       Figure 1 is hard to understand. It deserves a real legend, and some arrows needs to be explained.

-       I’m not sure that Table 1 help understanding the whole picture of the discussed mechanisms. It would be preferable to replace this table by a figure, so the reader could figure out activators and cell targets.

Author Response

Responses to reviewer #1

  1. The role of peroxisome proliferation and peroxisomal fatty acid beta-oxidation are well known to play an important role in thermogenesis. Knowing that peroxisomal fatty acid oxidation produce heat and not ATP and this pathway is dependent on nutritional conditions, authors could also discuss this point.

Answer: The related statement and references about peroxisome proliferation and peroxisomal fatty acid beta-oxidation have been added in 6.3. (line: 418-420, page:11) as suggested.

  1. Several abbreviations are not explained: UCP1, F-FDG-PET/CT..etc

Answer: Revised as suggested.

  1. Figure 1 is hard to understand. It deserves a real legend, and some arrows needs to be explained.

Answer: Revised as suggested. Accordingly, figure legend has been substantially rewritten and the explanation of BAT activation and DIT in health and diseases has been added.

  1. I’m not sure that Table 1 help understanding the whole picture of the discussed mechanisms. It would be preferable to replace this table by a figure, so the reader could figure out activators and cell targets.

Answer: Thanks for the reviewer’s comment. The purpose of table 1 is to summarize the updated animal and human studies focusing on BAT activation and DIT and the effect on total EE and energy balance. Accordingly, the headnote and the content of table 1 have been substantially revised to clarify this issue.

Reviewer 2 Report

The paper by Pei-Chi Chan and Po-Shiuan Hsieh ‘The role and regulatory mechanism of brown adipose tissue activation in diet-induced thermogenesis in health and diseases’ is a narrative review of the potential role of brown adipose tissue (BAT) activation and diet-induced thermogenesis (DIT) in weight management and metabolic disorders associated with obesity. The review addresses various aspects of this broad field of research, some in-depth others not. In my opinion, some additions and insights can help to improve the quality of the work:

·       Thyroid hormones play a central role in the control of thermogenesis and energy balance; however, the authors did not consider the contribution of these hormones. In addition, the authors analyzed the potential contribution of leptin (section 5.5), but not of its counterpart, adiponectin.

·       Pag 3 starting from line 142: “Recent studies have demonstrated that multiple UCP1 independent thermogenic mechanisms exist…”. It should be better explained how these UCP1 independent mechanisms work.

·       Pag 6 starting from line 240: mitochondrial glycerophosphate oxidase activity is particularly elevated in BAT. The role of this biochemical process in thermogenesis should be deepened

·       Pag 7 starting from line 286: Do these caloric intakes refer to studies conducted on humans or animals?

·       Section 6.3: retinoic acid and a vitamin A metabolite are not usually considered fatty acids. Moreover, both diets low in vitamin A and diets too rich in vitamin A are associated with body weight gain and adiposity

·       Both NAFLD and dyslipidemia characterized by high plasma triglycerides and low plasma HDL-cholesterol (thus is not only hyperlipemia), are associated with reduced insulin sensitivity and/or peripheral resistance to insulin. This should be considered in the paragraphs devoted to NFALD and this dyslipidemia.

Author Response

Responses to reviewer #2

  1. Thyroid hormones play a central role in the control of thermogenesis and energy balance; however, the authors did not consider the contribution of these hormones. In addition, the authors analyzed the potential contribution of leptin (section 5.5), but not of its counterpart, adiponectin.

Answer: Thanks for reviewer’s comments. The related statement and references about the potential contribution of thyroid hormones and adiponectin in DIT have been added in 5.7. (line: 325-334, page: 9) and 5.5. (line: 302-307, page: 8) as suggested.

  1. Pag 3 starting from line 142: “Recent studies have demonstrated that multiple UCP1 independent thermogenic mechanisms exist…”. It should be better explained how these UCP1 independent mechanisms work.

Answer: Revised as suggested in 4.2. (line: 178-183, page: 5).

  1. Pag 6 starting from line 240: mitochondrial glycerophosphate oxidase activity is particularly elevated in BAT. The role of this biochemical process in thermogenesis should be deepened.

Answer: Thanks for the comment from the reviewer. However, we cautiously rechecked the related statement mentioned by the reviewer. It actually showed that mitochondrial glycerophosphate oxidase activity is enhanced in liver instead of BAT after cafeteria feeding as shown in 5.4. (line: 279-282, page: 8). The underlying mechanism of this hepatic enzyme activation and the causal link with BAT activation remains unknown.

  1. Pag 7 starting from line 286: Do these caloric intakes refer to studies conducted on humans or animals?

Answer: The related statement has been further clarified and revised in 6.1. (line: 346-349, page: 10).

  1. Section 6.3: retinoic acid and a vitamin A metabolite are not usually considered fatty acids. Moreover, both diets low in vitamin A and diets too rich in vitamin A are associated with body weight gain and adiposity

Answer: Thanks for the comments from the reviewer. Accordingly, the related statement has been substantially revised and corrected. In addition, the studies about the dietary vitamin A supplement or deficiency with body weight gain and adiposity have been extensively addressed in 6.4. (line: 438-441, page: 12).

  1. Both NAFLD and dyslipidemia characterized by high plasma triglycerides and low plasma HDL-cholesterol (thus is not only hyperlipemia), are associated with reduced insulin sensitivity and/or peripheral resistance to insulin. This should be considered in the paragraphs devoted to NFALD and this dyslipidemia.

Answer: The suggestion from the reviewer is well-taken. Accordingly, the related discussions have been added in 7.3 and 7.4. (line: 508-511 and 516-519, page: 14).

Round 2

Reviewer 2 Report

I consider the revised version of the manuscript suitable for publication in Nutrients.